# Is Greener Better? Quantifying the Impact of a Nature Walk on Stress Reduction Using HRV and Saliva Cortisol Biomarkers

**DOI:** 10.3390/ijerph21111491

**Published:** 2024-11-09

**Authors:** Shravan G. Aras, J. Ray Runyon, Josh B. Kazman, Julian F. Thayer, Esther M. Sternberg, Patricia A. Deuster

**Affiliations:** 1Center for Biomedical Informatics and Biostatistics, University of Arizona, Tucson, AZ 85721, USA; 2Andrew Weil Center for Integrative Medicine, College of Medicine, University of Arizona, Tucson, AZ 85721, USA; jrayrunyon@arizona.edu (J.R.R.); esternberg@arizona.edu (E.M.S.); 3Department of Environmental Science, University of Arizona, Tucson, AZ 85721, USA; 4Consortium for Health and Military Performance, Department of Military and Emergency Medicine, F. Edward Hébert School of Medicine, Uniformed Services University of the Health Sciences, Bethesda, MD 20814, USA; josh.kazman.ctr@usuhs.edu (J.B.K.); patricia.deuster@usuhs.edu (P.A.D.); 5Henry M. Jackson Foundation for the Advancement of Military Medicine, Inc., Bethesda, MD 20814, USA; 6Department of Psychological Science, School of Social Ecology, University of California, Irvine, CA 92697, USA; jfthayer@uci.edu; 7Department of Medicine, College of Medicine, University of Arizona, Tucson, AZ 85724, USA; 8With Joint Appointments in Department of Psychology, College of Science, University of Arizona, Tucson, AZ 85721, USA; 9Architecture, Planning & Landscape Architecture, Nutritional Sciences and Wellness, College of Agriculture and Life Sciences, University of Arizona, Tucson, AZ 85721, USA; 10BIO5 Institute, University of Arizona, Tucson, AZ 85721, USA

**Keywords:** heart rate variability, saliva cortisol biomarkers, Green Road, urban road, mood disturbances, physiological stress

## Abstract

The physiological impact of walking in nature was quantified via continuous heart rate variability (HRV), pre- and post-walk saliva cortisol measures, and self-reported mood and mindfulness scores for N = 17 participants who walked “The Green Road” at Walter Reed National Military Medical Center in Bethesda, Maryland. For N = 15 of the participants, HRV analysis revealed two main groups: group one individuals had a 104% increase (mean) in the root mean square standard deviation (RMSSD) and a 47% increase (mean) in the standard deviation of NN values (SDNN), indicating an overall reduction in physiological stress from walking the Green Road, and group two individuals had a decrease (mean) of 42% and 31% in these respective HRV metrics, signaling an increase in physiological stresses. Post-walk self-reported scores for vigor and mood disturbance were more robust for the Green Road than for a comparable urban road corridor and showed that a higher HRV during the walk was associated with improved overall mood. Saliva cortisol was lower after taking a walk for all participants, and it showed that walking the Green Road elicited a significantly larger reduction in cortisol of 53%, on average, when compared with 37% of walking along an urban road. It was also observed that the order in which individuals walked the Green Road and urban road also impacted their cortisol responses, with those walking the urban road before the Green Road showing a substantial reduction in cortisol, suggesting a possible attenuation effect of walking the Green Road first. These findings provide quantitative data demonstrating the stress-reducing effects of being in nature, thus supporting the health benefit value of providing access to nature more broadly in many settings.

## 1. Introduction

There is substantial literature indicating that forests and other locations with natural green features have the potential to induce calm, restoration, and recovery [1,2,3,4]. These findings have motivated the creation of “green healing spaces” in locations where they are typically lacking, such as in urban centers or hospitals. Such efforts are important in several settings, including within the US Department of Defense (DoD), where this current study, which quantifies the physiological impact of a purposely designed nature walk to reduce stress and promote healing, was carried out at the Walter Reed National Military Medical Center [5,6]. Positive subjective responses to this nature walk, termed the “Green Road”, were reported using both self-report mood scales and qualitative interviews [7]. Measures of heart rate variability (HRV) and cortisol are shown to quantitatively capture the positive impact of walking the Green Road compared to an urban road on both the neuronal and hormonal stress response and are summarized herein.

Green healing spaces are inviting, satisfying, and desirable, but city and facility planners face several tradeoffs and considerations when trying to maintain or allocate funding for such resources. Despite a growing body of research documenting enhanced mood in response to walking in green areas, the physiological mechanisms are less well-characterized [3,8,9]. Obtaining quantitative data informing the mechanisms by which such spaces promote healing will provide additional support for the implementation of such green spaces as adjuncts for improving health. Two mechanisms that have been proposed to mediate the beneficial effects of nature include impacts on autonomic nervous system (ANS) responses to natural elements [8] and processes that refresh and enhance attention [10,11]. Calming effects are detected through relative increases in parasympathetic nervous system (PNS) activity and decreases in sympathetic nervous system (SNS) activity (e.g., slower heart rate and breathing). The measurement of heart rate variability (HRV) is one way to assess ANS activity directly in a non-invasive and continuous manner with minimal participant burden [12,13,14]. HRV captures the interactions between the SNS and PNS cores of the ANS, with a sympathetic outcome leading to an increased HR due to stressors and parasympathetic related to reduced HR during periods of rest [15].

Several studies in the past have shown an association between various HRV metrics and stress. A significant decrease was observed in the RMSSD (root mean square standard deviation) and SDNN (standard deviation of NN values) time domain HRV metrics in university students before an examination vs during, indicating an association with stress conditions [15,16]. The RMSSD captures short-term changes between successive RR beats, which correlated with changes to the PNS core, while the SDNN captures longer variability trends correlated to both SNS and PNS cores. As it is not always feasible to collect long-term 24 h ECG recordings for HRV analysis in mobile settings, several studies have experimented with short-term (less than 60 s) ECG recordings. A study using ultra-short-term HRV recordings showed a significant difference between rest and stressors (Stroop color test) for RMSSD and pNN50 (Percentage of NN intervals over 50 ms) metrics when using a window size of 30 s [17]. A comprehensive study on various HRV metrics versus the minimum window for identifying significant changes between a rest and a TSST (Trier Social Stress Test) found a window size of 50 s for calculating time domain metrics—the RMSSD, SDNN, and AVNN (Average value of NN intervals) were found to be significant biomarkers for detecting changes in mental stress [18]. However, only a few studies have examined HRV in response to environmental manipulations, and these have provided mixed evidence related to changes in PNS activity. In three laboratory studies, subjects were exposed to different virtual environments after a cognitive or social stressor. In two of the studies, which tested the environments using a virtual reality headset [19] and wrap-around movie screen [20], changes in HRV did not significantly vary based on the virtual environment. Notably, in the third study, which tested four virtual environments using computer images, HRV (LF/HF ratio) was significantly lower in response to the artificial environment (“square space”) compared to the three natural/green environments (“natural open”, “under forest”, “tree-line linear”) [21]. As the authors note, this effect might have been due to either the relaxing effect of the natural conditions or the non-relaxing effect of the control condition (“square space”).

Four field experiments examined whether walking in a forest or green area increases HRV (High Frequency) relative to walking in an urban environment. In three of the studies, participants were immersed in a natural or urban environment for a brief amount of time (~30 min). In two of them—both from Japan—the natural environment was associated with increases in HRV, consistent with reduced stress and an enhanced relaxation response [22,23]. In the third study, from Denmark, both environments elicited similar increases in HRV, although the control condition was a historic town and, therefore, might have had its own calming effect [24]. By examining real locations, these studies likely increased their external validity; however, they did include lengthy transportation or baseline periods, likely simulating a mild stressor. In a more recent study, subjects walked once weekly for three weeks in one of two conditions—a green area or a suburban area [25]. Following a two-week washout, they then switched conditions. Like the previous studies, this study documented statistically significant higher HRV in the green compared to the suburban condition, again suggesting reduced stress and enhanced relaxation response. Reasons for the mixed responses in the published studies include variability in methods, differences in designs, and the populations studied (cultural variation).

With the work presented here, we evaluate the role of using HRV metrics to examine changes in PNS activity between participants walking on an urban road compared to walking on the Green Road. Our primary objective was to calculate HRV metrics continuously during both walks to determine if subjects experienced suppressed SNS activity and an increase in the PNS response, equating to an overall increase in the SDNN and RMSSD (with a higher increase in RMSSD) for the green walk compared to the urban walk. The collected HRV stress response data are supplemented with salivary cortisol measures to measure the hormonal stress response. Forest bathing for at least 15 min has been shown to significantly reduce cortisol levels when compared with an urban environment [26,27]. In our study herein, the combination of HRV, cortisol, and subjective survey data provides a holistic assessment of the impact of the two different environments on both arms of the stress response.

## 2. Materials and Methods

### 2.1. Location

In 2017, “The Green Road” was built at Naval Support Activity, Bethesda (NSA/B), to promote healing among service members based on previous research findings that walking on greener roads can help reduce stress. Walter Reed National Military Medical Center (Walter Reed), the DoD’s largest treatment facility, is located on NSA/B, where many short- and long-term military patients, their family members, active duty and civilian staff, and medical students reside, visit, or work. The Green Road consists of a two-acre garden located in an eight-acre woodland ravine surrounding an existing natural stream.

### 2.2. Inclusion and Exclusion Criteria

The study was approved by the Uniformed Services University’s Institutional Review Board. We recruited 20 subjects between September 2018 and November 2019. Inclusion criteria included being a service member, military retiree, military family member, caregiver, student, or NSA/B employee; being ambulatory; aged 18–60 years; able to speak and read English; able to complete study questionnaires; able to abstain from food, alcohol, and tobacco for one hour prior to experiment; lack of overt heart disease or diagnoses limiting mobility; not pregnant or lactating.

### 2.3. Experimental Procedures

The Green Road shown in Figure 1, a 1.2-mile woodland path, is one of the US’ largest wild-type healing gardens and was completed in September 2017. This two-acre garden is situated in an eight-acre woodland ravine with an existing natural stream. In 2017, the woodland was modified to enhance exposure to natural elements (trees, water, stones, and wild animals) based on elements deemed desirable by focus groups of military personnel. The Green Road includes a Communal and Commemorative Pavilion (see Supplement for additional detail), while additional features, such as journaling benches, provide invitations for reflection. The road was built as a public/private partnership between the NSA/B and the Institute for Integrative Health (Baltimore) with substantial funding from TKF Foundation (Annapolis, MD, USA), the friends of Shockey Gillet, Capital Funding, LLC, and other donors.

The urban road shown in Figure 1 was comprised of concrete sidewalks and crosswalks on a relatively busy campus surrounded by buildings, multi-level parking garages, signs, small grassy areas, and some trees. The road has mostly man-made features, with some touches of nature. It borders on a memorial plaza featuring a Navy anchor, a cement plaque, and a tall statue in honor of Walter Reed.

After informed consent, all participants walked on both roads on different days with the order randomized. The two walks were frequently scheduled within a few days of each other to accommodate for schedules; the median number of days apart across subjects was four days (range: 1–13). All sessions started in the morning before 9 a.m. and included the following: body weight, height, and blood pressure measurements; completing demographics/trait questionnaires (first day only); placement of HRV monitor and chest electrodes; a five-minute pre-walk HRV recording while the subject was sitting down with legs and arms uncrossed; a collection of pre-walk salivary sample; completion of state questionnaires (see below); walking on the Green or urban road for approximately 20 min; a five-minute post-walk HRV recording similar to the pre-walk; post-walk saliva sample collection; post-walk state questionnaires; qualitative interview.

### 2.4. Physiological Measurement System and HRV Metrics Calculation

To calculate HRV metrics during the green and urban walks, participants were asked to wear a single lead ECG device called Bodyguard v2. Inter-beat interval (IBI) values obtained from this device were then used to calculate two time domain HRV metrics—the RMSSD and SDNN over a finite non-overlapping sliding window using custom-written Python code. Based on past literature on short-term HRV recordings, we used a window size of 60 s for both the RMSSD and SDNN [18]. A single HRV metric was calculated for each 60 s interval over the entire walk (both urban and green). This allowed us to study the changes in HRV trends as the walk progressed for both roads. Participants walked for exactly 20 min on the urban road and between 20 and 22 min on the Green Road, which gave rise to an uneven number of samples for both walks. We used spline interpolation during the walks to generate 20 HRV points (RMSSD and SDNN) each for every participant to simulate homogenous sampling, reduce motion artifacts, and account for missed samples from Bodyguard 2. A mean square error (MSE) was used to measure the fit error between the spline and raw HRV data, and a smoothing factor of *s =* 0.4 was selected for the fitted spline. Figure 2 shows an example of the smoothened spline fitted to the raw HRV metrics for participant GR2. Colored boxes indicate the durations of the active walk, with green indicating the Green Road and blue the urban road. A full figure that shows all participants can be found in Figure A1 for figures.

### 2.5. Psychological Measurements

We examined two self-report state scales—the 37-item Profile of Mood States (POMS) and the 5-item Mindfulness Attention Awareness Scale (MAAS) [28,29]. Both scales asked about how participants were feeling in the present moment and were administered four times—before and after the Green Road and the urban road. The POMS consists of 37 single-word adjectives (e.g., worn-out) and asks participants to rate how they are feeling right now along a five-point Likert-type scale ranging from “not at all” to “extremely”. Scores from the POMS include depression, anger, confusion, vigor, and tension, and a total mood disturbance score (based on all the subscales and with the vigor items reverse-coded). For these analyses, only the total mood and vigor subscale scores were included because the other scores had very strong floor effects (≥50% receiving the minimal score). These two scale scores demonstrated strong reliability at all timepoints, with an alpha of 0.78 for total mood and 0.94 for vigor.

The MAAS has five items asking about how much attention is directed to present circumstances (e.g., “I was rushing through something without being really attentive to it”). Participants rate each item along a seven-point Likert-type scale with anchors ranging from “not at all” to “very much”. Within the current study, the post-walk reliability was 0.75 for the urban road and 0.85 for the Green Road.

### 2.6. Statistical Analysis

Analyses were primarily descriptive and intended to depict dynamic changes in HRV between walking on the Green Road versus the urban road. Additional exploratory analyses examined associations between HRV-derived metrics and state scales. HRV metrics throughout the walk were aggregated for both roads, and a pairwise *t*-test was used to calculate the *p*-values to quantify significant differences in physiological stress indications.

### 2.7. Raw Data and Analysis Code

Raw IBI values from Bodyguard 2, along with the self-reported state-scales data, are available on request. All source code used for analysis in this work is organized in Python Jupyter notebooks and publicly available on GitHub—https://github.com/UArizonaCB2/GreenroadHRV.git, accessed on 27 September 2024.

### 2.8. Saliva Cortisol Measurements

Cortisol was measured in salvia samples collected from each participant before and after their walk using an established liquid chromatography-mass spectrometry method which is summarized [30].

The LC system consisted of a Vanquish +UPLC coupled with a Thermo Scientific TSQ Quantiva triple quadrupole mass spectrometer. A Restek Raptor column (2.7 um particle diameter, 2.1 × 100 mm column dimensions) was used to achieve retention of cortisol in standard solutions and saliva samples with binary gradient elution. Mobile phase A consisted of 0.1% formic acid in LCMS water, and mobile phase B consisted of 0.1% formic acid in LCMS acetonitrile. The flow rate was maintained at 300 μL/min with the following gradient: 0–1 min hold 0% B: from 1 to 2 min ramp from 0% to 20% B: from 2 to 10 min ramp from 20 to 25% B: from 10 to 11 ramp from 25 to 100% B and hold for 1 min; from 12 to 12.5 min ramp down from 100 to 0% B and hold for 2.5 min to prepare the column for the next injection. The total run time for the method was 15 min. The sample vials were maintained at 10 °C in the autosampler, and the column was maintained at 37 °C. Ionization of cortisol was achieved using atmospheric pressure chemical ionization (APCI) in positive ion mode. The ion transfer tube and vaporizer temperatures were 350 °C and 450 °C, respectively. The sheath gas and axillary gas were set to 30 and 1 arbitrary units. The APCI probe discharge current was set to 2 μA. The mass resolution was set at 1.2 to increase the sensitivity by allowing the detection of more ions near the target molecular weight, especially since baseline resolution was achieved in the LC step.

Cortisol was tracked using MS/MS selective reaction monitoring with a mass transition of 363.2 > 121.1 m/z and a collision energy of 23 eV. Cortisol concentrations in saliva samples were calculated from the peak area of the mass transition peak using a calibration curve generated from cortisol standard (Steraloids, Inc.; Newport, RI, USA) solutions designed to capture endogenous concentrations ranging from 0.2 to 50.0 ng/mL. Nine calibration solutions were prepared via 2x serial dilution of 50 ng/mL cortisol with 5 ng/mL deuterated cortisol (Cortisol-9,11,11,12-D4; CDN Isotopes Inc.; Pointe-Claire, QC, Canada) added as an internal standard in each sample. Linear least-squared regression with a weighting factor of 1/(x^2^) was used for the curve fitting of the standards, excellent linearity was observed with an equation of y = 784.39x, and the intercept passing through the origin (R^2^ = 0.9919). LCMS water with 5 ng/mL internal standard was used as a blank. A total of 10 μL of each calibration solution and each saliva sample were injected into the LC-MS/MS for analysis. Saliva samples for LC-MS/MS analysis were prepared by centrifuging the raw saliva sample at 16,000× *g* rcf through an Amicon 0.5 mL 10 kDa molecular weight cutoff regenerated cellulose membrane and then taking 1 part saliva filtrate and mixing it with 4 parts LCMS water to create a 5x diluted sample.

### 2.9. Retrospective Stratification of Participants into Groups Based on HRV Changes

To differentiate among participants who showed a positive change between walking the Green Road versus the urban road, participants were retrospectively stratified into groups based on those that showed a positive impact, negative impact, and mixed impact on all HRV metrics. The HRV change was calculated as [*HRV_Green_ − HRV_Urban_*] for both RMSSD and SDNN metrics. A positive change was defined by an increase in both HRV metrics when walking the Green Road, which was indicative of a reduced physiological response, and these participants were assigned to group 1. On the other hand, those that showed a negative change were defined by a decrease in both HRV metrics, which was indicative of an increased physiological response, and these participants were assigned to group 2. Finally, those who showed an increase in one metric and a decrease in the other, indicating a different rate in SNS and PNS changes, were assigned to group 3.

## 3. Results

One of the biggest challenges with quantifying stress responses is collectively accounting for individual variabilities across three different and often independent axes—autonomic nervous system (ANS) stress responses as measured by HRV, responses from the hormonal axis, as measured by salivary cortisol markers, and psychological effects measured using self-reported scales. To highlight both similarities and dissimilarities between these axes, we report the results separately for each domain metric (HRV, self-reported mood scales, and salivary cortisol) and compare changes between HRV and cortisol observed for our participants individually and in aggregate throughout the study.

### 3.1. Study Population Demographics

In this study, a total of 20 participants walked both the Green Road and the urban road. Three participants (one male and two females) were removed from analyses due to corrupted raw RR values from the Bodyguard v2 ECG device worn by the participants, which prevented us from calculating HRV metrics. This availed an analytic sample of 17, with a median age of 34 years old, an age range from 19 to 60 (interquartile range: 27, 44), and mostly female (82%) participants. Over a third were Caucasian (35%), followed by African American (24%), Asian American/Pacific Islander (24%), bi-racial (12%), and American Native (6%). The body mass index (BMI) was between 25 and 30 for 21% of females and 67% of males and over 30 for 21% of females and no males.

### 3.2. Heart Rate Variability

Table 1 presents HRV and cortisol data obtained during the Green Road and urban road walks for each participant. Delta (Δ) HRV values show the difference in the mean RMSSD and SDNN between green and urban walks, respectively. A positive delta value with a *p* < 0.01 indicates a significant increase in that HRV metric on the green as compared to the urban walk. These differences have been indicated with a (**). We see three different patterns in our participants: (1) those showing an increase in both HRV metrics on the green as compared to the urban road (reduced physiological stress); (2) those showing a decrease in both HRV metrics on the green as compared to the urban road (non-reduction in physiological stress); (3) those showing increases in one of the HRV metrics for the Green Road (inconclusive indication of physiological stress change). We indicate these participants with (1), (2), and (3), respectively, with 52% in 1 (N = 9), 35% in 2 (N = 6), and 11% in 3 (N = 2).

### 3.3. Cortisol

The percent change in measured salivary cortisol after taking a walk is shown in the two right columns in Table 1. Cortisol was lower after taking a walk for all participants independent of the walking path. However, walking the Green Road elicited a significantly larger reduction in cortisol (53%) compared to walking the urban road (37%), as shown in Figure 3a.

We observed an average decrease in cortisol of 53% when walking the Green Road compared to 37% for the urban road. Further, as shown in Figure 3b, we observed an impact on measured cortisol that was dependent on which road was walked first. A significant decrease in measured cortisol was found when participants walked the Green Road after walking the urban road (an average decrease of 68%). By comparison, an increase in measured cortisol was observed for participants who walked the urban road after the Green Road (an average increase of 9%).

### 3.4. Comparing Physiological Stress Interpretation Between HRV and Cortisol Metrics

Table 2 summarizes the changes in HRV and cortisol metrics resulting from walking the Green Road compared to the urban road for each retrospectively assigned group. The difference between both metrics was measured as (green−urban) for each participant’s mean and then aggregated to calculate the overall mean (μ) and standard deviation (σ). ∆% for all three metrics was calculated as [μ(green) − μ(urban)) × 100/μ(urban)]. For all individuals in group one, we saw a 104% mean increase in the RMSSD and a 47% mean increase in SDNN values, indicating an overall reduction in physiological stress, as indicated by HRV measurements. This was complemented by our findings on cortisol, which showed a mean decrease of 56% for walking on the Green Road compared to walking on the urban road.

On the other hand, we saw a decrease of 42% and 31% in the respective HRV metrics for those in group two, signaling a possible increase in physiological stresses, as measured via HRV metrics alone. However, for the group two individuals, we also saw a mean decrease in cortisol of 27%, indicating a contradictory narrative of reduced stress responses when using hormonal markers. While the overall decrease in cortisol for group two was lesser than that for group one individuals (27% vs. 56%), group two is important in highlighting the independent nature of the physiological response via HRV and hormonal responses via cortisol and their combined use for overall stress evaluation.

For the remaining two individuals in group three, we saw a mean decrease of 40% in cortisol, a very small increase of only 1.8% for the RMSSD, and a mean decrease of −0.2% for the SDNN. In this case, HRV metrics for the RMSSD align well with a reduction in cortisol and indicate a reduced stress response; however, SDNN metrics offer a contradicting viewpoint.

### 3.5. Self-Report Scales

We examined partial correlations between HRV metrics reported in Table 1 and self-reported scales (total mood disturbance, vigor, mindfulness) (Table 3). The correlations are adjusted for pre-walk measures of HRV and pre-walk mood scores. Given the small sample and exploratory nature, we omit *p*-values and focus on the general pattern. Compared to post-walk resting HRV, HRV measures taken during the walks had a larger association with post-walk self-report scores, particularly for vigor and mood disturbance; these associations were more robust for the Green Road than for the urban road. Higher scores indicate increased vigor and worse moods. For the mood scales (total mood and vigor), the associations were in the expected directions, with a higher HRV being associated with an improved overall mood. Associations between HRV and state-mindfulness were less consistent and for RMSSD were in the opposite direction than expected.

## 4. Discussion

This study has shown that taking a walk on the Green Road has a significant positive impact on the stress response compared to walking an urban path, even for the limited number of participants. The combination of physiological (HRV) and hormonal (cortisol) measures provides a holistic approach to capturing the impact of walking a Green Road on stress responses compared to either measure alone. Furthermore, the employment of continuous HRV processing using sequential short-term time windows (60 s) across the total time of the walk is a novel data analysis approach that yields a detailed analysis of the dynamic sympathetic and parasympathetic responses during the walk.

Most of the participants in this study showed physiological (HRV—RMSSD and SDNN) responses, and all participants showed hormonal responses (cortisol), which support the positive impact of walking the Green Road on stress reduction. HRV metrics capture the interaction between the sympathetic and parasympathetic nervous systems, where the SDNN represents the overall autonomic function (SNS + PNS) and the RMSSD reflects the parasympathetic modulation [31,32,33]. Cortisol release is mediated through stimulation of the hypothalamic–pituitary–adrenal (HPA) axis, which is activated via the SNS. An increase in the RMSSD relative to the SDNN would indicate preferential activation of the PNS, which, in turn, would downregulate the HPA axis, thus reducing the measured cortisol.

Grouping the individuals retrospectively based on changes in their HRV metrics allowed the relative effects of both the Green Road and urban road environments on the participants to be quantified. For group one (reduced physiological stress), we saw a significant within-subject increases in both HRV metrics for the participants, with a larger mean increase in RMSSD (almost double) values compared to SDNN. For these individuals, we argue that walking on the Green Road showed a marked increase in their PNS activity relative to their SNS activity and a concurrent decrease in cortisol of 56% on average (Table 3).

For group two (non-reduction in physiological stress), we saw significant within-subject decreases in both HRV metrics for the participants when walking on the Green Road compared to the urban road. A larger decrease of the RMSSD (mean 42%) compared to the SDNN (mean 31%) suggests that these individuals had an elevated SNS activity relative to PNS activity when walking the urban road compared to the Green Road. A decrease in cortisol of 27% on average (Table 3) was observed in group two, suggesting the influence of the PNS on the inhibition of the HPA axis, albeit to a smaller extent compared to group one, because of the competitive relative increase in SNS activity with this group.

All participants showed a larger decrease in cortisol after walking the Green Road compared with the urban road (Figure 3a), highlighting the positive impact of the Green Road on reducing physiological stress compared with the urban road. The order in which individuals walked the Green Road and urban road also impacted their cortisol response (Figure 3b). Thus, those walking the urban road before the Green Road showed a substantial reduction in cortisol, while those walking the Green Road before the urban road showed a small elevation of cortisol while walking the urban road. These results highlight the influence of the relative RMSSD-to-SDNN response to each walk while also suggesting a possible attenuation from the first walk. Habituation and intensity of a stressor have been shown to attenuate a subsequent stressor, even several days later, where eventually, the stress response to a known stressor may become indistinguishable from a placebo effect [34,35,36]. For example, if a participant walks the Green Road first, they may anticipate a similar stress load when walking the urban road. The resulting decrease in the RMSSD and SDNN relative to the expectation would result in elevated cortisol relative to the Green Road. Vice versa, if a participant walks the urban road first, they may experience an increase in the RMSSD and SDNN from expectation, thus resulting in lower cortisol.

Cortisol levels are not expected to increase during or after low-intensity exercise [34,37]. Salivary cortisol was significantly reduced when a person was exposed for at least 15 min to a forest environment compared with an urban environment or even viewing a forest landscape [26,27]. Our results are in agreement with these findings. The reduction in cortisol after a low-intensity walk along the urban road supports the stress-reducing effect of simply taking a walk. The larger observed cortisol reduction (double) after taking a walk on the Green Road highlights the enhanced stress-reducing impact of walking in nature/forests. Indeed, guided mindful walks have been shown to reduce stress and anxiety [38]. The ANS response, or relative PNS/SNS, is a rapid response to stress, which, in turn, can up- or downregulate the hormonal response via the HPA axis accordingly to yield a systemic stress response. Our data capture these dynamically different and independent controls of the two arms of the stress response—the physiological (HRV, ANS) and hormonal (cortisol/hypothalamic pituitary adrenal axis)—to provide a comprehensive quantitative assessment of the stress response.

The participants exhibited a diverse range of individual differences in HRV metric deltas, ranging from a minor change for GR3 (R = 1.7, S = 3.93) to a relatively large change for GR17 (R = 120.25, S = 95.58). This wide spread of HRV changes suggests that while walking on the Green Road had a beneficial impact on reducing stress, there is a high degree of individual variability that can stem from baseline physiological conditioning and overall experiences during the walk. Appendix A Figure A1 shows a plot of HRV metrics for all individuals in group one, further highlighting the individual differences. We noticed that for all participants, except for GR2 and GR14, HRV metrics dropped during the walk and came back to the baseline once the walk was over. On the other hand, participants such as GR17 showed a remarkable increase in HRV metrics prior to the Green Road, as compared to almost no change prior to (or during) the urban walk. We also note that participants had a wide range of baseline HRVs, measured during a 5 min window prior to the walk when participants sat still without moving or talking, with their legs and ankles uncrossed. During the urban walk for this group, the baseline RMSSD ranged from 13 ms (GR20) to 70 ms (GR5), with a mean of 40 ms and a standard deviation of 20 ms. On the other hand, the baseline SDNN for this group ranged from 33 ms (GR20) to 99 ms (GR15), with a mean of 65 ms and std of 24 ms. For the Green Road, the baseline for the RMSSD ranged between 16 ms (GR20) and 124 ms (GR5), with a mean of 53 ms and a std of 34 ms. The SDNN ranged from 46 ms (GR3) to 111 ms (GR5), with a mean of 75 ms and a std of 23 ms. In both urban road and green road walk days (walks happened on different days), the mean baseline for the SDNN was higher in comparison to the RMSSD, indicating the individuals had a much higher SNS activation compared to the PNS prior to both walks. When comparing the baseline between the two days, we saw a mean difference of 17 ms (std = 16 ms) for the RMSSD and 17 ms (std = 9 ms) for the SDNN. This further highlights the variety in individual HRV characteristics on days when the two walks were conducted. We also note that GR8 (group 2) reported seeing a snake on Green Road, further increasing the challenge of controlling such factors on an open walk. Additionally, the difference in terrain, such as the elevation grade and surrounding noises (GR11 reported being disturbed by construction noises in urban), could not be controlled for such real-world tests. Taken together, these findings are consistent with the stress-attenuating effects of nature exposure; in this case, walking the “Green Road”. However, there was considerable individual variability in autonomic nervous system (ANS) stress responses as measured by HRV, with some individuals showing a substantial positive impact on HRV and others little or none.

## 5. Limitations

While our current study has shown the importance of green spaces for stress reduction, we acknowledge the use of a limited sample size of 17 participants who were included in this study. However, this limited sample size was still large enough to capture the independent differences between HRV and cortisol for interpreting stress reduction. We believe this study paves the way for future diverse studies, which can use multiple variables to compensate for reduced study sizes.

We also had several incidences where participants were affected by environmental factors, such as seeing a snake on the Green Road, construction noise, and uphill climbs. As one of our objectives was to study the effects of green spaces in the wild (non-lab environment), we cannot fully eliminate these disturbances. Some ways to control for such disturbances would be to drop participants who reported experiencing such external disturbances, provide a bigger sample size, or repeat the experiment at a different time. However, owing to these environmental disturbances, our study closely mimics laboratory and real-world results.

Several previous studies have highlighted the individualized variations in HRV markers related to physical fitness, previous day’s activity workload, recovery, sleep, exercise, age, smoking, etc., further making the use of HRV markers challenging [39,40,41,42]. While it is not always possible to account for all baseline and background variations, participants in a healthy BMI range were enrolled in our study. They were asked to refrain from certain activities prior to participation to minimize the impacts of these extraneous variables on the results. Participants were asked to start the walk before 9 a.m. Furthermore, we also explored the possibility of the BMI or age being a differentiating factor for HRV variation across groups one, two, and three; however, we were unable to find any.

Finally, cortisol was shown to decrease for all participants regardless of the walk. This may partially be impacted by the timing of the experiment (before 9 a.m.). In future studies, the experiment should be repeated in the afternoon as well as the morning, similar to other exercise-based studies [43], to better isolate the Green Road impact versus possible diurnal variations in the observed cortisol [44].

## 6. Conclusions

The unique combination of measured physiological and hormonal responses to walking the Green Road was very individualistic among the participants and highlighted the independent nature of autonomic nervous system (ANS) stress responses (continuous HRVs), responses from the hormonal axis (salivary cortisol markers) and psychological effects (self-reported mood scales). We saw an overall decrease in cortisol of 53% and 37% for all participants after walking both the Green Road and urban road, respectively. This suggested that walking on any road reduced stress; however, walking on a Green Road had a larger impact on stress reduction because of the increased PNS response. HRV results were mixed, with 53% of the participants showing an increase in both RMSSD and SDNN metrics, indicating a reduction in stress; 35% of the participants showed an increase in both metrics, indicating no stress reduction, and the remaining 11% showed an increase in only one metric, indicating a mixed response to stress reduction. Additionally, we also found a larger association between during- and post-walk self-report scores, particularly for vigor and mood disturbance, with these associations being more robust for the Green Road than for the urban road. This suggests that future studies into autonomic-mediated improvements in mood states should focus more on measuring HRV during an activity or in the moment rather than after a change that is already presumed to have occurred (e.g., during a manipulation, not post-manipulation). These findings provide quantitative data, demonstrating the stress-reducing effects of being in nature and supporting the health benefit value of providing access to nature more broadly in many settings. Our current study makes a positive argument for the purposeful inclusion of biophilia and green spaces of different sizes, such as gardens, parks, and woodlands in urban environments, and spaces for stress reduction [45,46] and paves the way for future large sample size studies to quantify the effects of stress reduction in these environments.

## Figures and Tables

**Figure 1 ijerph-21-01491-f001:**
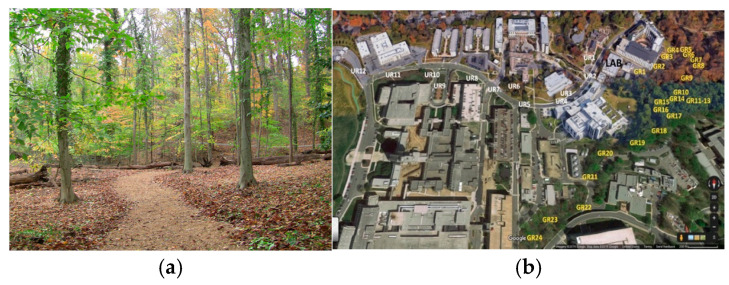
Images of the Green Road and the urban road walked by participants. (**a**) The Green Road was a 1.2-mile woodland pathway and (**b**) the urban road comprised of concrete sidewalks and crosswalks.

**Figure 2 ijerph-21-01491-f002:**
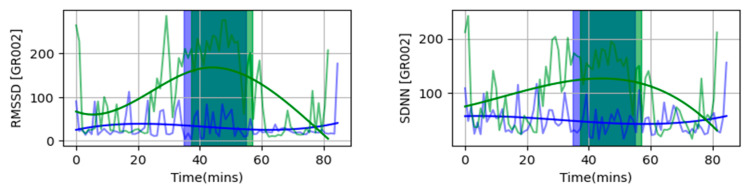
Examples of a smoothen spline fitted to raw HRV metrics for GR2. The urban road is indicated by blue and the Green Road by green. Boxes show durations of walk.

**Figure 3 ijerph-21-01491-f003:**
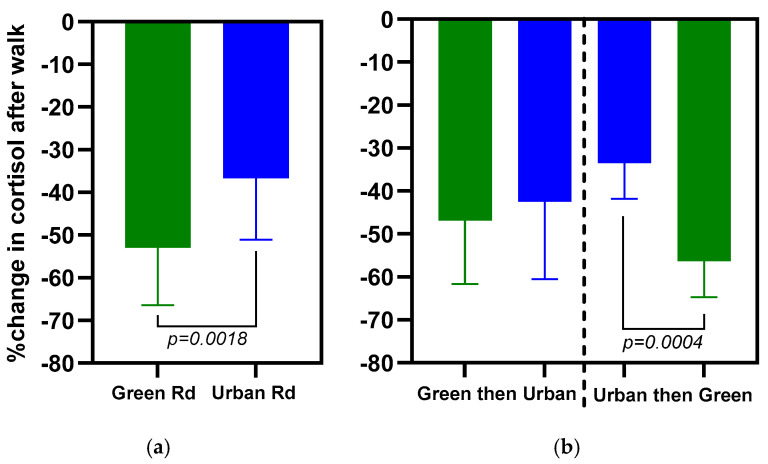
Results from saliva cortisol meassurements. (**a**) Walking the Green Road resulted in a significantly larger reduction in cortisol compared with walking the urban road. However, walking either path resulted in decreased cortisol. (**b**). The cortisol response was dependent on the sequence the paths were walked. The data in Figure 3 is an aggregate of all participants across the groups.

**Table 1 ijerph-21-01491-t001:** Summary of RMSSD, SDNN and cortisol measures compared between the Green Road and urban road for individual participants.

ID	RMSSD (ms)		SDNN (ms)		Δ HRV (ms)		Δ Cortisol (%)	
	Urban	Green	Urban	Green	RMSSD	SDNN	Urban	Green
Group 1 (Increase in both HRV metrics during Green Road as compared to urban road walk) N = 9								
GR2	33 ± 4 (25–39)	117 ± 41 (60–167)	50 ± 4 (42–56)	108 ± 17 (74–126)	83 **	57 **	−18.8	−46.6
GR3	21 ± 11 (11–48)	22 ± 10 (14–47)	37 ± 10 (24–50)	41 ± 6 (35–53)	1.7 **	3.93 *	−24.7	−47.9
GR5	41 ± 29 (16–112)	56 ± 41 (15–143)	49 ± 23 (27–101)	62 ± 34 (30–135)	15 **	14 **	−58.2	−59.2
GR14	36 ± 15 (15–55)	39 ± 10 (29–65)	58 ± 17 (35–79)	60 ± 6 (55–78)	2.97	2.58	−25.8	−70.9
GR15	40 ± 15 (24–70)	57 ± 12 (33–71)	61 ± 20 (43–106)	67 ± 9 (48–78)	17 **	7	−28.9	−67.9
GR16	22 ± 11 (13–49)	30 ± 8 (20–41)	45 ± 14 (29–69)	53 ± 14 (36–78)	8 **	8 **	−59.3	−59.9
GR17	45 ± 12 (34–76)	165 ± 66 (13–235)	72 ± 9 (64–93)	168 ± 57 (13–227)	120 **	96 **	−10.0	−42.7
GR18	19 ± 5 (14–29)	37 ± 12 (15–51)	35 ± 7 (28–50)	43 ± 9 (26–53)	18 **	7 **	−50.4	−46.7
GR20	19 ± 4 (15–28)	20 ± 11 (10–46)	35 ± 4 (30–44)	40 ± 11 (30–67)	0.22	4	−41.2	−60.1
Group 2 (Decrease in both HRV metrics during Green Road as compared to urban road walk) N = 6								
GR1	129 ± 13 (113–158)	77 ± 25 (36–106)	120 ± 13 (110–153)	85 ± 17 (55–106)	−52 **	−34 **	−19.5	−47.4
GR6	72 ± 30 (44–140)	56 ± 24 (33–108)	82 ± 20 (58–113)	73 ± 16 (59–110)	−16 **	−9 **	−46.4	−68.4
GR7	88 ± 16 (64–110)	61 ± 11 (43–75)	87 ± 17 (61–110)	66 ± 13 (45–83)	−27 **	−21 **	−43.7	−21.2
GR8	108 ± 42 (61–185)	63 ± 22 (30–90)	122 ± 39 (77–189)	86 ± 17 (63–106)	−46 **	−37 **	−50.5	−58.1
GR11	50 ± 32 (24–134)	25 ± 9 (12–35)	62 ± 30 (38–142)	33 ± 4 (27–39)	−26 *	−30 **	−29.6	−49.7
GR13	42 ± 5 (36–53)	14 ± 9 (4–34)	44 ± 2 (42–50)	25 ± 7 (18–39)	−28 **	−19 **	−40.6	−47.4
Group 3 (Mixed increase or decrease in HRV metrics) N = 2								
GR10	49 ± 11 (32–62)	52 ± 10 (35–63)	71 ± 18 (49–101)	69 ± 3 (65–74)	3	−2	−44.6	−71.4
GR19	83 ± 34 (52–169)	80 ± 22 (61–136)	106 ± 24 (82–162)	108 ± 17 (92–145)	−2	2	−31.6	−35.4

For each HRV metric the mean is calculated across all 60 s windows during the walk. Additionally it also shows the percentage change in cortisol between start and end of the walks (Δ Cortisol). We report the standard deviation, and range for the windowed metrics for HRV. The difference indicates green–urban metrics. Significance. Codes for HRV: ** *p* < 0.01, * *p* < 0.1.

**Table 2 ijerph-21-01491-t002:** Summary of the impact of walking the Green Road compared to the urban road on HRV metrics (RMSSD, SDNN) and cortisol for groups 1, 2 and 3.

Groups	∆ RMSSD (ms)			∆ SDNN (ms)			∆ Cortisol (% Decrease Before and After Walk)		
	μ	σ	∆%	μ	σ	∆%	μ	σ	∆%
1	37.5	42	104%	27.5	32.6	47%	−20	20	56%
2	−31.6	10.9	−42%	−24.9	9.6	−31%	−10	18	27%
3	0.44	2.57	1.8%	0.18	1.96	−0.2%	−15.3	16	40%

μ calculated as an average of individual differences within a group (green−urban). Δ represents the percent increase or decrease in HRV and cortisol metrics for Green Road averaged throughout the group as compared to urban road. ∆% = [μ(green) − μ(urban)) × 100/μ(urban)].

**Table 3 ijerph-21-01491-t003:** Partial correlations between heart rate variability, mood states, and mindfulness during and after walking the Green Road and the urban road.

	RMSSD				SDNN			
	During		Post		During		Post	
	Green	Urban	Green	Urban	Green	Urban	Green	Urban
Moods								
Vigor	0.48	0.22	0.17	−0.13	0.45	0.31	0.22	−0.16
Total mood disturbance	−0.49	−0.25	−0.21	0.11	−0.48	−0.36	−0.27	0.13
Mindfulness Attention	0.17	−0.34	−0.40	−0.29	0.15	−0.17	−0.07	−0.22

Note. Moods are from POMS [28] and mindfulness from the MAAS [29]. Both scales query about states in the moment. For all of the mood constructs, except for vigor, higher scores indicate worse moods. Partial correlations examined post-walk levels of each scale, adjusting for pre-walk levels and baseline HRV (e.g., the correlation between vigor and RMSDD in the Green Road adjusted for pre-walk vigor and pre-walk RMSSD on that day).

## Data Availability

The original contributions presented in the study are included in the article, further inquiries can be directed to the corresponding author/s.

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
