# Peer review of "Is Greener Better? Quantifying the Impact of a Nature Walk on Stress Reduction Using HRV and Saliva Cortisol Biomarkers"

_ijerph, 2024, doi:10.3390/ijerph21111491_

Round 1
Reviewer 1 Report
Comments and Suggestions for Authors
The study aimed to quantify the physiological impact of walking in nature versus an urban environment by measuring continuous heart rate variability (HRV), pre- and post-walk saliva cortisol levels, and self-reported mood and mindfulness scores for 17 participants. The researchers found that participants could be divided into three groups based on their HRV response: those who showed an increase in both HRV metrics (RMSSD and SDNN) during the green road walk, those who showed a decrease, and those with a mixed response. The green road walk was associated with a 104% increase in RMSSD and a 47% increase in SDNN on average for the first group, indicating reduced physiological stress. Saliva cortisol levels also showed a larger reduction after walking the green road compared to the urban road. The study provides novel quantitative data demonstrating the stress-reducing effects of spending time in nature. However, the high individual variability in responses suggests the need for further discussion and analysis to better understand the factors influencing the physiological outcomes. Revision of the manuscript is needed to improve the depth and clarity of the findings and their implications.
1- What potential mechanisms could explain the observed individual differences in HRV responses between the three identified groups during the green road walk? Factors such as baseline autonomic balance, physical fitness, and psychological traits should be considered. More discussion is needed on how these factors may interact to produce the varied HRV outcomes.
2- Since the order of walking the green and urban roads impacted the cortisol response, what potential physiological or psychological explanations could account for this sequencing effect? Investigating potential carry-over or priming effects between the two environments would provide valuable insights.
3- Given the small sample size, how can the authors enhance the statistical power and generalizability of their findings? Suggestions may include expanding the sample, replicating the study in different populations, or employing meta-analytic techniques to synthesize results with prior studies in this area.
4- The authors mention that terrain differences and external stimuli (e.g., seeing a snake) may have influenced the HRV responses. How could the study design be improved to better control or account for these environmental factors that could impact the physiological outcomes? More discussion is needed on strategies to minimize confounding variables.
5- Expand the discussion on the potential mechanisms underlying the individual differences in HRV responses, including the role of baseline autonomic balance, physical fitness, and psychological traits. Explore how these factors may interact to shape the varied physiological outcomes.
6- Provide a more in-depth analysis and discussion of the sequencing effect observed for the cortisol response. Investigate potential carry-over or priming effects between the green and urban environments and their implications for understanding the stress-reducing benefits of nature exposure.
7- Discuss strategies to enhance the statistical power and generalizability of the findings, such as expanding the sample size, replicating the study in different populations, or employing meta-analytic techniques to synthesize the results with prior studies in this area.
8- Elaborate on the potential confounding effects of environmental factors, such as differences in terrain and external stimuli, and describe methods to better control or account for these variables in future research. Discuss the implications of these uncontrolled factors on the interpretation of the physiological outcomes.
9- Provide a more detailed comparison of the study's findings with relevant literature in the field, highlighting both the similarities and differences in the observed physiological responses to nature exposure.
10- Strengthen the discussion on the practical implications of the study's findings, particularly in the context of designing and implementing green healing spaces in various settings, such as healthcare facilities and urban centers.
11- Consider restructuring the introduction to provide a more comprehensive and cohesive review of the existing literature on the physiological effects of nature exposure, including both laboratory and field-based studies. This would help contextualize the current study within the broader research landscape.
12- Improve the organization and flow of the results section, potentially by presenting the HRV, cortisol, and self-report findings in a more integrated manner, rather than as separate sub-sections. This could help the reader better understand the interplay between the different physiological and psychological measures.
Comments on the Quality of English LanguageIt needs minor revision.
Reviewer 2 Report
Comments and Suggestions for Authors
Abstract: good
Background: well done with appropriate citations.
Methods:
2.1 line129-134 should be taken out or revised to eliminate the reference to their previous publication. The content can be included without the phrasing related to “Previously published our results…”
2.2 line143-150 should be moved to the results section under demographics and titled “study population.” The section 2.2 should be retitled “inclusion and exclusion criteria.”
2.4 line 196 “Error! Reference source not found.2,” please explain or resolve.
2.5 line 217 Related to the POMS: Why are the “alphas around?” please put in accurate alphas for these two items.
2.5 line 221-222 Are these the reliability statistics from the present study? Please be specific.
2.6 were the comparisons “between” or “across” the events? This section is unclear and should be detailed further.
Line 242 again there is reference to previous publication and this need to be details rather than referenced for the audience.
Lines 243-247 did you also stratify the cohort who walked on the urban road?
Results
Should start with study population demographics.
Table 1 significant codes: there are no noted *** significances and this should be removed from the caption.
Table 2 Groups. What do the groups refer to when using asterisks? These are not helpful. Overall, the narrative and Table 2 are confusing. Is this the entire sample, green and urban? Can you clarify?
Cortisol results are simpler to read. Please be obvious about what Group 1 and Group 2 represent throughout.
Discussion
There are some detailed data that were collected from the participants prior to the intervention that should be detailed in the methods section. Ex: crossed ankles
Conclusion for further studies using HRV during is a key finding/recommendation.
Reviewer 3 Report
Comments and Suggestions for Authors
Both HRV and health supporting effects of nature experiences are current topics of interest.
background adequate and steps taken to minimize variability in nature tx exp vs urban road
excellent use of outcome measures to provide convergent validity when there are many factors which can influence results. Settings, procedures and measures described in detail which can be a challenge in nature-based research. Mixed methods is a good choice here
A little more detail of subject recruitment process and rationale of determination of subject number (power test, etc) would improve this section
line 196 clarify or remove error statement
limitations of this study and nature-based research adequately addressed
line 387 should 'work' be 'walk'?
Round 2
Reviewer 1 Report
Comments and Suggestions for Authors
The revised version of the manuscript can be considered for publication.